# Trends and Inequalities in the Health Care and Hypertension Outcomes in China, 2011 to 2015

**DOI:** 10.3390/ijerph16224578

**Published:** 2019-11-19

**Authors:** Yang Zhao, Ajay Singh Mahal, Tilahun Nigatu Haregu, Ameera Katar, Brian Oldenburg, Luwen Zhang

**Affiliations:** 1The Nossal Institute for Global Health, The University of Melbourne, Melbourne 3010, Australia; zhao11@student.unimelb.edu.au (Y.Z.); ajay.mahal@unimelb.edu.au (A.S.M.); haregu.t@unimelb.edu.au (T.N.H.); ameera.katar@unimelb.edu.au (A.K.); brian.oldenburg@unimelb.edu.au (B.O.); 2WHO Collaborating Centre on Implementation Research for Prevention & Control of NCDs, Melbourne 3010, Australia; 3School of Health Services Management, Southern Medical University, Guangzhou 500000, Guangdong, China

**Keywords:** socioeconomic inequality, healthcare, hypertension, trends, China

## Abstract

*Background:* Hypertension is one of the most significant and common risk factors for cardiovascular disease, yet it remains poorly controlled in China. This study aims to examine trends and socioeconomic inequalities in the management of hypertension between 2011 and 2015 in China and to investigate the association between antihypertensive medication treatment and reduction of blood pressure, using nationally representative data. *Methods:* Concentration curve and concentration index were used to assess socioeconomic-related inequalities in hypertension care and health service utilisation. The fixed-effects analysis was performed to measure the impact of medication treatment on reduction of blood pressure among people with hypertension by using linear regression models. *Results*: Among hypertensive individuals, there were growing trends in the rates of awareness and treatment from 55.87% and 48.44% in 2011, to 68.31% in 2013 and 61.97% in 2015, respectively. The proportion of hypertension control was still below 30%. The fixed-effects models indicated that medication treatment was statistically significant and associated with the patients’ systolic blood pressure (β: −13.483; 95% CI: −15.672, −11.293) and diastolic blood pressure (β: −5.367; 95% CI: −6.390, −4.344). *Conclusions:* China has made good progress in the hypertension diagnosis, medication treatment and coverage of health services over the last 10 years; however, pro-rich inequalities in hypertension care still exist, and there is considerable progress to be made in the prevention, treatment and effective control of hypertension.

## 1. Introduction

In the era of sustainable development goals (SDG), advancing universal health coverage (UHC) has become the centrepiece of health policy in many countries [1]. In this context, one of the key goals is to reduce premature mortality from noncommunicable diseases (NCDs) by one-third by 2030, mainly through effective prevention and treatment [2]. Like most countries in the world, China has recently experienced a very rapid epidemiological transition from a preponderance of infectious diseases to NCDs, including hypertension, diabetes, cardiovascular disease and cancers [3]. In an attempt at an effective response, and to achieve UHC, the central government in China launched comprehensive health reform in 2009 to increase the population’s access to health services, and to lower the financial burden of illness [4].

Previous studies reported the degree of diagnosis, treatment and control of hypertension in China [5,6], but most of them are cross-sectional in design [7], not well suited, therefore, to capturing changes in treatment adherence and blood pressure control over time [8]. There are relatively few longitudinal studies assessing trends in hypertension management and health service utilisation over the last 10 years, since health system reform in China [9]. To our knowledge, this is the first panel analysis in China that employed fixed-effects models at the national level to examine the impact of hypertension treatment adherence on blood pressure control.

In this study, trends and socioeconomic inequalities were evaluated in healthcare and control of hypertension between 2011 and 2015 in China. Additionally, the association between antihypertensive medication treatment and reduction of blood pressure, using a nationally longitudinal dataset, was investigated, controlling for changes in treatment status for different periods.

## 2. Materials and Methods

### 2.1. Data Source

Data from baseline and follow-up surveys in the China Health and Retirement Longitudinal Study (CHARLS) undertaken in 2011, 2013 and 2015 was utilised. This study aimed to collect a nationally representative sample of Chinese residents aged 45 and over to inform scientific research on the elderly and to assess health trends. The CHARLS design, based on the Health and Retirement Study (HRS) and other related ageing surveys, has received funding support from Peking University, the Behavioural and Social Research Division of the National Institute on Aging, and the World Bank [10].

The CHARLS’ questionnaires cover the following domains: demographics, health status and functioning, healthcare and insurance, income and consumption, and several important biomarkers including height, weight and blood pressure. In CHARLS, each respondent's systolic blood pressure (SBP) and diastolic blood pressure (DBP) were recorded three times by a trained nurse using a HEM-7112 electronic monitor. The average value for each study participant was calculated, but only given to the subjects once the interviews were completed. The interviewees were asked if they had hypertension and whether they were receiving any form of antihypertensive treatment. Details on how blood pressure measurements were taken in the CHARLS survey can be found elsewhere [10].

To ensure a representative sample, the CHARLS baseline survey covered 150 counties/districts and 450 villages/urban communities, across 28 provinces, by using multistage stratified probability-proportionate-to-size (PPS) sampling. A total of 17,708 individuals in 10,257 households were successfully interviewed for the baseline CHARLS survey. The response rate was over 80% in all age-eligible households. Ongoing follow-up surveys were conducted every two years [10].

After excluding cases with missing demographic information and blood pressure measurements, complete data were available for 13,725 individuals in 2011, 10,893 individuals in 2013, and 11,675 individuals in 2015. The three rounds of the CHARLS datasets were used to calculate the prevalence of hypertension and healthcare indicators for the different years. To measure the impact of medication treatment on blood pressure (BP), this research also created a panel dataset including 8,486 hypertensive individuals who could be identified in all rounds of the CHARLS by excluding the loss to follow-up and the deceased in 2013 and 2015.

### 2.2. Indicators

In this study, hypertension was defined as survey participants having systolic blood pressure ≥140 mmHg and/or diastolic blood pressure ≥90 mmHg, and/or their being on antihypertensive medication treatment for raised blood pressure [11,12]. Among hypertensive adults identified, those whose systolic blood pressure was less than 140 mmHg and those whose diastolic blood pressure was less than 90 mmHg were considered as effectively controlling their hypertension [13]. The measure of health service utilisation included three variables: the number of outpatient visits, the number of inpatient visits, and the number of education sessions provided by health professionals. The definition of prevalence, awareness, treatment and control of hypertension, and the variables in health service use are listed in Table 1.

### 2.3. Statistical Analysis

The prevalence of hypertension and rates of awareness, treatment and control amongst Chinese adults were calculated. The Chi-square trend test was used to evaluate the difference in trends in the prevalence of hypertension, healthcare utilisation and blood pressure control from 2011 to 2015. Concentration Curve (CC) and Concentration Index (CI) were used to assess socioeconomic-related inequalities in hypertension care and service use. The further the CC lies from the equality line (45-degree line), the higher the degree of inequality in healthcare and health service utilisation [14].

The concentration index can quantitatively reflect the degree of equality. This study calculated ***CI*** to measure socioeconomic inequalities in healthcare and control for hypertension according to the following equation:*CI* = 2/*μcov*(*h*,*r*)(1)
where *h* refers to the variable in healthcare; ***μ*** is the mean of ***h***; and ***r*** refers to the fractional rank of people according to the per capita household consumption expenditure (PCE), from the poorest to richest. A positive ***CI*** value shows that healthcare or service use is more likely in those in a higher socioeconomic group, and a negative ***CI*** value shows that underprivileged people are more likely to use health services or to be controlled well. Similar to the concentration curve, the higher the absolute value of ***CI***, the worse for those with lower socioeconomic status. Zero indicates no socioeconomic inequality in hypertension care [14].

Given that treatment adherence and the within-person blood pressure can easily change over time, fixed-effects models (FEM) were performed to measure the impact of medication treatment on the reduction of blood pressure among those with hypertension by using linear regression analysis. The patients’ SBP and DBP were included in regression models as outcome variables. The explanatory variables of interest were age, per capita household consumption expenditure, Body Mass Index (BMI), comorbidity, current smoking and alcohol consumption. Correlates of hypertension treatment were analysed using the FEM, after controlling for demographic characteristics. In addition to estimating the main effects of the treatment on blood pressure, interaction effects were examined by introducing the interaction term of treatment and some core variables, including PCE, BMI and comorbidity. Considering the potential systematic error caused by sampling design and missing values, both the weighted and unweighted proportions for key variables of hypertension management were reported in descriptive results, using the weighting codes created by the CHARLS team [10]. All statistical analyses were conducted using STATA 14.0 (Stata Corp, College Station, TX, USA). *p* values less than 0.05 were considered statistically significant.

## 3. Results

### 3.1. Trends in Hypertension Prevalence and Healthcare in China

Overall, the prevalence of hypertension among adults in China aged 45 and above was approximately 40.78% in 2011, 43.25% in 2013 and 41.81% in 2015 (Table 2). Among patients who identified as having hypertension, there were growing trends in the rate of awareness and treatment from 55.87% and 48.44% in 2011 to 68.31% and 61.97% in 2015, respectively. The level of hypertension control was still low in China during the years 2011 to 2015. Specifically, the hypertensive patients with complications were more likely to be aware of their hypertension status, receive antihypertensive medications and to have achieved blood pressure control, compared to those without complications (Table 2).

For the utilisation of health services, Table 3 shows that inpatient care use increased from 10.68% in 2011 to 17.41% in 2015 in China. There was a growing number of hypertensive individuals receiving health education from health professionals from 2011 (33.67%) to 2013 (36.55%) and 2015 (39.79%). Patients with complications were also using more health services, compared to those without any complications; however, we did not find a significant change in the number of outpatient visits by adults in China with hypertension between 2011 and 2015.

### 3.2. Socioeconomic Inequalities in Healthcare and Outcomes

The results of socioeconomic-related inequality analyses for healthcare among patients with hypertension are shown in Table 4. The proportions of medication treatment and control in the wealthiest quintile of people with hypertension were 1.16 times and 1.41 times greater, respectively, than in the most deprived quintile in China in 2015. Meanwhile, the proportion of outpatient visits, inpatient visits and health education sessions in the wealthiest quintile of hypertensive patients were 1.47, 2.91 and 1.36 times greater, respectively, than in the most deprived quintile.

The concentration curves for hypertension care were below the equality line, indicating that hypertension care and the use of health services were disproportionately higher among wealthy individuals in China in 2015. In terms of hypertension management, the worst indicator of pro-rich inequality was the control of blood pressure (Figure 1). Regarding the health service use among hypertensive patients, compared with the outpatient service and health education access, the most unequal indicator was the inpatient visit, which was also more concentrated among wealthy individuals (Figure 2).The concentration index of inpatient visits was 0.086, the only indicator with a statistically significant difference. In terms of health education, there was a slight concentration among poorer individuals (Table 4).

### 3.3. Results of Multivariate Regression Analysis

Table 5 shows the main results of the fixed-effects model analysis. The fixed-effects models indicated that medication treatment was statistically significant and associated with the patients’ systolic blood pressure (β: −13.483; 95% CI: −15.672, −11.293) and diastolic blood pressure (β: −5.367; 95% CI: −6.390, −4.344), before introducing the interaction term of treatment and characteristic variables into these models. After introducing the interaction terms of antihypertensive medication treatment with PCE, BMI and comorbidity, all fixed-effects models continued to show statistical significance. There was, however, no significant association between interaction terms and a reduction in patients’ blood pressure, as Model 2 showed.

## 4. Discussion

This study demonstrates that the utilisation of consistent and regular medical treatment has improved the effective control over individuals with hypertension over time. This pattern is likely to reflect the successful implementation of recent policies and the intended outcomes of new health system reform which aim to achieve universal health coverage with affordable and equitable primary healthcare [15,16]. Since 2009, China’s health system reform has focused on five major areas, including improvements in the primary healthcare delivery system, public health insurance schemes, the essential medicine system, implementation of the National Public Health Initiative (NHPI) and public hospital improvements (pilot reform in 16 cities). The standard and qualified management of hypertension are directly linked to this comprehensive reform [17].

### 4.1. Trends in Hypertension Prevalence and Treatment in China

This study showed a relatively high and stable prevalence of hypertension among middle-aged and elderly Chinese, with a downward turning point appearing in 2015. Meanwhile, it also reported an increased rate of hypertension awareness, treatment and control. The effective management of hypertension among the middle-aged and elderly is generally attributed to the implementation of the NPHI, which started in 2009 and aims to offer equitable essential public health services for all Chinese people, including hypertension and diabetes screening, standard management and health education [18].

At the same time, the essential medicine system (EMS) was also rolled out across the country. This aims to improve medication access, quality and appropriate use, and this policy has benefited the majority of patients with hypertension, diabetes and some other NCDs [19,20]. Through the EMS, primary care facilities in townships and counties in China have access to more than 400 essential medicines, including various antihypertensive drugs (such as beta-blockers and calcium channel blockers) [21]. Significantly, the provision of these medicines is heavily subsidised by the Chinese government through the implementation of zero-profit mark-up for essential medicines. The previously exploited 15% price mark-up was banned [22]. With the development of the EMS, and with more medication being reimbursed under the insurance schemes in China, this reform could lead to substantial improvements in hypertension treatment in the near future.

### 4.2. Health Service Use among Chinese Adults with Hypertension

Overall, a substantial upward trend in health service utilisation was observed among Chinese adults with hypertension between 2011 and 2015. The new health system reform in China played a significant role in promoting this trend [17]. The national programme of primary healthcare, for instance, spends around 6.3 billion US dollars every year on community healthcare providers who deliver a defined package of healthcare, including management of NCDs and essential primary healthcare [15]. They are responsible for establishing a health record and providing free health examinations and health education, including for people with hypertension.

Improvements in health insurance policies over the past few years have also played a positive role in hypertension management. The health system reform resulted in a significant increase in insurance coverage, with 95.7% of the Chinese population being covered by public health insurance schemes in 2011 [23]. Along with the expansion of benefits packages from public insurance schemes, financial risk protection has been improved in China. This may partially explain the findings of a growing trend in inpatient care. The utilisation of outpatient services, however, did not increase between 2011 and 2015, in large part because the New Rural Cooperative Medical Scheme, in which almost all rural residents in China were enrolled, did not cover the cost of outpatient care [17].

The experiences of several developing and developed countries have demonstrated the relationship between insurance and healthcare [24,25,26,27,28,29,30,31,32]. Financial protection factors could play an important role in the improvement of health service utilisation for hypertension. Hypertensive patients insured through Mexico’s Seguro Popular, for instance, are highly likely to receive antihypertensive treatment [24]. Among American adults with hypertension, those without health insurance had lower rates of blood pressure control between 2005 and 2008 [33]. A large-scale trial in the United States also showed that demand-side reimbursement from health insurance could increase primary and preventive services use [34].

### 4.3. Low Level of Control for Hypertension in China

This study revealed that the blood pressure of hypertensive patients was higher in people without medication than among those who reported the use of antihypertensive medication. Although the performance of hypertension management has been reported in China on several occasions [35,36,37,38,39,40,41,42,43], very few studies have conducted a longitudinal analysis in the same sampling group [5], and none of them employed the fixed-effects models. The FEMs in this study suggested that medication treatment plays a vital role in blood pressure control among people with hypertension; however, the overall degree of controlled hypertension was universally low across three survey waves from 2011 to 2015. The rate of BP control among hypertensive Chinese adults was less than 20% across the nation, compared with 48% in the United States [33].

The reality of two-thirds of hypertensive patients experiencing uncontrolled blood pressure is likely to be the result of inconsistent, inadequate treatment or insufficient quality of health service provision. Though the NPHI was developed to ensure optimal management of hypertension and diabetes for Chinese adults aged 35 and above, the inconsistent and uncoordinated primary care delivery system, lack of qualified physicians and public health workers and absence of effective surveillance system handicapped the implementation of the NPHI [44]. A survey showed that over 60% of Chinese adults with hypertension were still not receiving public health services in 2015, that is, there was a lack of health education to the population, including advice on weight control, exercise, healthy diet and smoking control. Major changes need to be made to improve the efficiency of service delivery for hypertension and substantially improve the effective control of hypertension in China. In response to this challenge, the current primary care system in China needs to be further strengthened and integrated with other reforms in public health service provision and the EMS. With comprehensive and continuous efforts in reforming the healthcare system, a more efficient and equitable health system would result, ensuring further improvement in the management of chronic diseases in China.

### 4.4. Socioeconomic Inequalities in Hypertenison Care

This study suggested levels of hypertension awareness, treatment and control tended to be higher among wealthier individuals in China, while a similar pro-rich distribution of health service use was observed, particularly the inpatient visit. The economic-related inequalities were consistent with previous findings regarding hypertension detection and treatment in China [45,46], Brazil [47] and Tanzania [48]. The socioeconomic inequality in hypertension control was pro-rich in Malaysia and South Africa, and several other low- and middle-income countries, although it was relatively equitable in Iran due to good access to affordable medicines [49]. These findings still highlight the importance of ensuring universal access to affordable, efficient and appropriate public health programs to detect hypertension in the middle-aged and elderly, and when diagnosed, to provide continuous access to essential treatment, including medications.

A good degree of hypertension care and control is unlikely to be achieved unless the universal healthcare and public insurance coverage can be further strengthened to improve access to both primary care and essential medications for all [50]. In 2019, China’s central government rolled out relevant policies to ease burdens on citizens living with chronic disease. Hypertensive patients covered by urban and rural resident health insurance will be reimbursed for outpatient medications on the EMS, with the reimbursement ratio to be raised to over 50% [51]. Future research is needed to further evaluate the effectiveness of reforming these health insurance policies on hypertension management in China.

### 4.5. Strengths and Limitations

This is the first study in China to apply fixed-effects models in examining the impact of antihypertensive medication on reductions in blood pressure, considering the change of treatment status in different periods. This research will contribute to a deeper understanding of reform policies related to primary healthcare and public health services, and further analysis of the effectiveness of NCD treatment. This study only considered the middle-aged and older population in China, however. The effectiveness of hypertension treatment for younger Chinese groups should be considered in future studies. Some 20% of respondents were missing information on blood pressure during the surveys from 2011 to 2015. To account for nonresponse bias, we adjusted the analysis by using the created weights for individuals.

## 5. Conclusions

China has made good progress in the identification of people with hypertension, in medication treatment and in coverage of health services over the last 10 years. Between 2011 and 2015, China also saw a significant increase in the use of inpatient care services and access to health education related to lifestyle behaviours among hypertensive patients. This pattern reflects the implementation of recent policies. Despite these advances, fewer than a third of Chinese adults with hypertension were in control of their blood pressure, and pro-rich inequalities in the hypertension care and service use exist. China still needs to make more progress with respect to prevention, and to ensure more effective treatment and control of hypertension.

## Figures and Tables

**Figure 1 ijerph-16-04578-f001:**
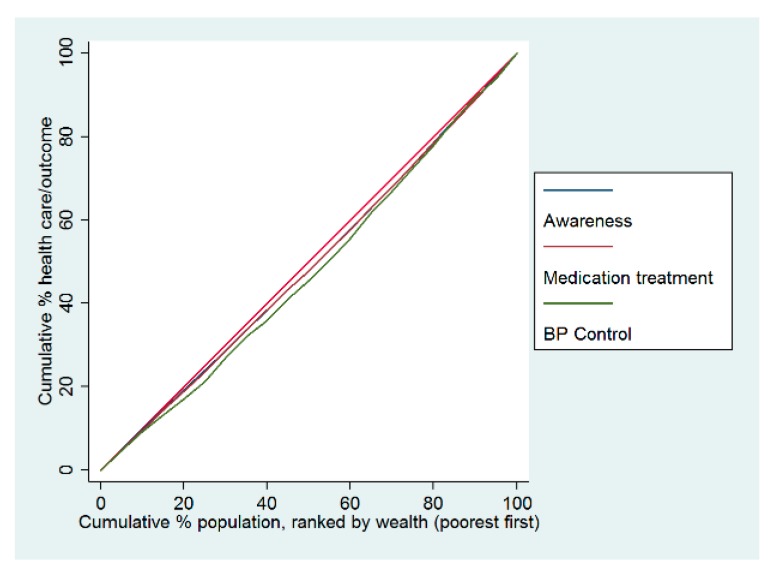
Concentration curves of healthcare for adults with hypertension in China in 2015.

**Figure 2 ijerph-16-04578-f002:**
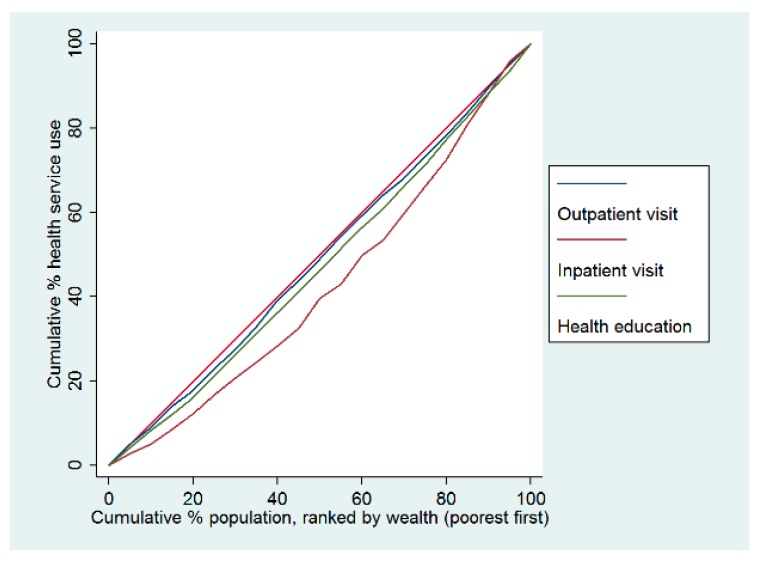
Concentration curves of health service use for adults with hypertension in China in 2015.

**Table 1 ijerph-16-04578-t001:** Definition of indicators for evaluating the management of hypertension.

Measures	Definition
Prevalence	Systolic blood pressure ≥140 mmHg and/or diastolic blood pressure≥90 mmHg and/or self-reported antihypertensive medication use at the time of the investigation
Hypertension care	
Awareness	Hypertensive persons who reported the doctor's diagnosis of hypertension previously or current use of antihypertensive medication
Treatment	Identified hypertensive adults who reported current use of antihypertensive medication
Control	Identified hypertensive adults whose systolic blood pressure was <140 mmHg and whose diastolic blood pressure was <90 mmHg
Health service utilisation	
Outpatient visit	‘In the last month have you visited a public hospital, private hospital, public health centre, clinic, or doctor’s practice, or been visited by a doctor for outpatient care?’
Inpatient visit	‘Have you received inpatient care in the past year?’
Health education	Health education provided by health professionals, which includes weight control, exercise, healthy diet and aspects of smoking control

Note: In the calculation of hypertension prevalence, the denominator was the number of all respondents; in the calculation of the percentage of awareness, treatment and blood pressure control, the denominator was the number of all identified hypertensive patients.

**Table 2 ijerph-16-04578-t002:** Prevalence and healthcare of hypertension among adults 45 years and above in China from 2011 to 2015.

	2011			2013			2015			*p* Value
	N	Unweighted (%)	Weighted (%)	N	Unweighted (%)	Weighted (%)	N	Unweighted (%)	Weighted (%)	
Total population	13,725	100.00	100.00	10,893	100.00	100.00	11,675	100.00	100.00	
Prevalence	5285	38.51	40.78	4666	42.83	43.25	4874	41.75	41.81	<0.001
Among people with hypertension										
Awareness	2976	56.31	55.87	2876	61.64	60.04	3316	68.03	68.31	<0.001
Without comorbidity	755	43.07	42.19	598	46.39	44.48	734	54.29	55.80	<0.001
With comorbidity	2221	62.88	63.38	2278	67.46	66.77	2582	73.31	73.32	<0.001
Medication treatment	2579	48.80	48.44	2485	53.26	52.57	2980	61.14	61.97	<0.001
Without comorbidity	638	36.39	35.40	473	36.70	36.39	645	47.71	49.59	<0.001
With comorbidity	1941	54.95	55.60	2012	59.58	59.56	2335	66.30	66.93	<0.001
Blood pressure control	1072	20.28	19.34	1064	22.80	22.19	1326	27.21	28.57	<0.001
Without comorbidity	250	14.26	13.24	189	14.66	12.79	268	19.82	22.96	<0.001
With comorbidity	822	23.27	22.69	875	25.91	26.25	1058	30.04	30.81	<0.001

Note: Unweighted values referred to the results of calculating direct observations and weighted values referred to the results after adjusting the sampling design and nonresponding cases. *p* values are reported for intertemporal differences of the weighted proportion; in the calculation of the percentage of awareness, treatment and BP control, the denominator was the number of all identified hypertensive patients.

**Table 3 ijerph-16-04578-t003:** Trends in the health service use among people with hypertension by comorbidity status in China from 2011 to 2015.

Health Service Use	2011			2013			2015			*p* Value
	N	Unweighted (%)	Weighted (%)	N	Unweighted (%)	Weighted (%)	N	Unweighted (%)	Weighted (%)	
Proportion of outpatient visit	1029	19.72	21.24	1084	23.60	23.38	988	20.62	20.60%	0.105
Without comorbidity	200	11.53	13.65	166	12.98	13.16	179	13.44	14.21%	0.692
With comorbidity	829	23.81	25.43	918	27.70	27.84	809	23.39	23.18%	0.226
Proportion of inpatient visit	577	10.93	10.68	754	16.19	17.68	861	17.72	17.41%	<0.001
Without comorbidity	100	5.72	5.62	93	7.24	14.73	154	11.44	11.82%	<0.001
With comorbidity	477	13.51	13.45	661	19.60	18.95	707	20.12	19.66%	<0.001
Proportion of health education	1717	32.49	33.67	1713	36.71	36.55	1942	39.84	39.79%	<0.001
Without comorbidity	388	22.13	22.87	280	21.72	22.54	413	30.55	30.37%	<0.001
With comorbidity	1329	37.63	39.59	1433	42.43	42.60	1529	43.41	43.57%	<0.001

Note: Unweighted values referred to the results of calculating direct observations and weighted values referred to the results after adjusting the sampling design and nonresponding cases. *p* values are reported for intertemporal differences of the weighted proportion; health education provided by health professionals includes weight control, exercise, healthy diet and aspects of smoking control.

**Table 4 ijerph-16-04578-t004:** Healthcare for hypertensive adults in China across wealth quintiles and concentration indexes, 2015.

Wealth Group, PCE (Quintiles)	Awareness	Medication Treatment	BP Control	Outpatient Visit	Inpatient Visit	Health Education
1st (Poorest)	64.82%	57.71%	24.28%	17.41%	8.01%	31.63%
2nd (Poor)	65.43%	59.88%	27.99%	21.41%	14.28%	41.68%
3rd (Middle)	71.04%	63.52%	32.14%	17.84%	17.67%	46.70%
4th (Rich)	74.42%	65.07%	30.86%	18.86%	19.08%	47.20%
5th (Richest)	71.97%	66.66%	34.32%	25.65%	23.27%	42.95%
Ratio (Richest/Poorest)	1.11	1.16	1.41	1.47	2.91	1.36
CI^c^	0.007	0.010	0.008	0.033	0.086 ***	−0.004

Note: PCE, per capita household annual consumption expenditure; BP, blood pressure; Health education provided by health professionals includes weight control, exercise, healthy diet and aspects of smoking control. The proportions reported are weighted values. *** *p* < 0.001, ** *p* < 0.01, * *p* < 0.05 significance test.

**Table 5 ijerph-16-04578-t005:** Fixed-effects analysis of the relationship between antihypertensive medication treatment and blood pressure control.

Variable (Reference)	SBP-Model 1			SBP-Model 2			DBP-Model 1			DBP-Model 2		
	β	95% CI		β	95% CI		β	95% CI		β	95% CI	
Main effects												
Treatment (no)	−13.483	−15.672	−11.293	−16.115	−28.382	−3.848	−5.367	−6.390	−4.344	−8.326	−14.048	−2.604
Age	−0.386	−0.674	−0.097	−0.388	−0.677	−0.100	−0.121	−0.255	0.013	−0.121	−0.255	0.013
PCE, Quintile 1st												
2nd	−3.125	−5.419	−0.830	−1.736	−5.400	1.928	−0.508	−1.576	0.560	−0.002	−1.710	1.707
3rd	0.025	−2.338	2.388	2.094	−1.664	5.853	−0.215	−1.316	0.885	0.150	−1.604	1.903
4th	0.313	−2.179	2.806	2.757	−1.035	6.548	−0.231	−1.392	0.931	−0.441	−2.213	1.331
5th	−2.066	−4.740	0.608	−0.190	−4.574	4.193	−0.197	−1.442	1.047	0.402	−1.642	2.446
BMI	0.126	−0.221	0.474	0.005	−0.448	0.457	0.175	0.013	0.336	0.084	−0.126	0.295
Comorbidity (no)	−1.353	−4.980	2.275	−1.400	−5.839	3.039	−1.853	−3.545	−0.162	−1.596	−3.673	0.481
Smoking (no)	3.492	−0.586	7.569	3.327	−0.759	7.413	0.130	−1.773	2.032	0.133	−1.773	2.040
Drinking (no)	1.527	−1.084	4.139	1.512	−1.101	4.125	0.980	−0.236	2.196	0.975	−0.242	2.192
Interaction effects												
Treatment × PCE												
Treatment × 2nd				−2.381	−7.080	2.318				−0.815	−3.006	1.376
Treatment × 3rd				−3.459	−8.169	1.251				−0.629	−2.824	1.566
Treatment × 4th				−4.090	−8.879	0.698				0.353	−1.882	2.587
Treatment × 5th				−3.078	−8.323	2.167				−0.885	−3.329	1.559
Treatment × BMI				0.204	−0.262	0.670				0.146	−0.072	0.363
Treatment × Comorbidity				0.134	−4.080	4.348				−0.423	−2.396	1.550
Constant	176.336	156.896	195.776	178.044	157.463	198.625	91.426	82.380	100.471	93.207	83.630	102.784

Note: SBP, systolic blood pressure; DBP, diastolic blood pressure; PCE, per capita household annual consumption expenditure; BMI, body mass index; 95% CI, 95% confidence interval. All models were adjusted for age, living standard, BMI, comorbidity, smoking and drinking alcohol. Models 1 included only main-effects terms for all variables; Models 2 included both main-effects terms and interaction-effects terms for treatment and household consumption expenditure, BMI and comorbidity. × refers to the interaction term.

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
