# Peer review of "Trends and Inequalities in the Health Care and Hypertension Outcomes in China, 2011 to 2015"

_ijerph, 2019, doi:10.3390/ijerph16224578_

Round 1
Reviewer 1 Report
Zhao, et al. presented their article entitled, "Trends and inequalities in the health care and hypertension outcomes in China, 2011 to 2015 " that subjects with hypertension in China have good progress in medical treatment. Essentially, their information about populational surveillance about subjects with hypertension is important for both health provider and people in the country. Presentation of Table 2 to 4 seems to be inappropriately, therefore detailed information might be mis-understood by readers. This should be appropriately fixed. In abstract section, in line 3, between 2011 and 2013 might be mistyped. In Table 5, we couldnot find any statistical information about interaction effects between models, this should be presented appropriately.
Author Response
Dear reviewer,
We are extremely appreciated for your kind consideration of our manuscript (ijerph-611130), entitled "Trends and Inequalities in the Health Care and Hypertension Outcomes in China, 2011 to 2015". We are also grateful for your very helpful suggestions and constructive comments on our manuscript. We carefully considered every comment and made cautious revision accordingly. The following paragraphs (using the red colour) are our point-by-point responses to each of the comments. Please see the attachment.
If you have any other questions about this paper, we would quite appreciate it if you could let us know them at the earliest possible time.
Thanks a lot and with kind regards,
Dr Yang Zhao
Prof Ajay Singh Mahal
Prof Tilahun Nigatu Haregu
Ms Ameera Katar
Prof Brian Oldenburg
Dr Luwen Zhang
Oct 18, 2019.
Responses to the comments:
Zhao et al. presented their article entitled, "Trends and inequalities in the health care and hypertension outcomes in China, 2011 to 2015 " that subjects with hypertension in China have good progress in medical treatment. Essentially, their information about populational surveillance about subjects with hypertension is important for both health provider and people in the country.
Point 1: Presentation of Table 2 to 4 seems to be inappropriately; therefore, detailed information might be misunderstood by readers. This should be appropriately fixed.
Response 1: CHARLS set weighting code to the investigation samples to adjust the design of sampling and loss of follow-ups in the process of interviewing. One of the principal designers of CHARLS data, Yaohui Zhao, has compared the weighted demographic data for age-gender structure in CHARLS with the data from the Chinese population census in 2010 and found them comparable. Therefore, we also reported both the unweighted and weighted descriptions of demographic characteristics for readers to obtain a comprehensive understanding of the results.
We added the related explanation in the Method section, line 129-132 as follows:
“Considering the potential systematic error caused by sampling design and missing values. both the weighted and unweighted proportions for key variables of hypertension management were reported in descriptive results, using the weighting codes created by the CHARLS team.”
Meanwhile, we further improved the legend of the tables 2-3. The sentence for explaining “Unweighted values referred to the results of calculating direct observations and weighted values referred to the results after adjusting the sampling design and non-responding cases” were added the notes at the bottom of tables.
For the table 4, we aim to highlight the concentration indexes for principal indicators in this study, so we only keep the result of the weighted proportions as additional information for readers, which have better sampling representativeness than unweighted results. Sure, we add a caption for clarification as well. “The proportions reported are weighted values”
Point 2: In the abstract section, in line 3, between 2011 and 2013 might be mistyped.
Response 2: Thanks very much for pointing this out. We have revised it, by using “between 2011 and 2015” instead of “between 2011 and 2013”. (Line 16)
Point 3: In Table 5, we could not find any statistical information about interaction effects between models; this should be presented appropriately.
Response 3: we have added the statistical information about interaction effects between models used in this study, as the follows: “To estimating the main effects of the treatment on blood pressure, interaction effects were examined by introducing the interaction term of treatment and some core variables, including PCE, BMI, and comorbidity”. (Line 126-129)

Reviewer 2 Report
Many statements require adequate reference (f.i. Rows 44; 46; 48; 52; 212;233; 234)
Row 198 to 202 seem to belong more to the introduction rather than to the discussion.
Row 203 "reflects" implies that this is the only explanation, but this is not necessarily the case; better to use a more nuanced formulation such as "likely to reflect".
Some paragraph in discussion are not consequence of data analysis and could be dropped or made substantially synthetics (row 213-221; 274-286)
Table 2: the meaning of weighted and unweighted must be explicitly stated; also p values needs to be split for the two temporal intervals; similarly in table 3 ed and e captions need clarification
The graphic aspect of the figures1 and 2 can be improved
Author Response
Dear reviewer,
We are extremely appreciated for your kind consideration of our manuscript (ijerph-611130), entitled "Trends and Inequalities in the Health Care and Hypertension Outcomes in China, 2011 to 2015". We are also grateful for your very helpful suggestions and constructive comments on our manuscript. We carefully considered every comment and made cautious revision accordingly. The following paragraphs (using the red colour) are our point-by-point responses to each of the comments.
If you have any other questions about this paper, we would quite appreciate it if you could let us know them at the earliest possible time.
Thanks a lot and with kind regards,
Dr Yang Zhao
Prof Ajay Singh Mahal
Prof Tilahun Nigatu Haregu
Ms Ameera Katar
Prof Brian Oldenburg
Dr Luwen Zhang
Oct 18, 2019.
Responses to the comments:
Point 1: Many statements require adequate reference (f.i. Rows 44; 46; 48; 52; 212;233; 234)
Response 1: We have followed the reviewer’s advice and added adequate references / citations to these sentences mentioned.
Point 2: Row 198 to 202 seem to belong more to the introduction rather than to the discussion.
Response 2: Thanks for this very helpful suggestion. We have moved this part (Line 198-202) to the second paragraph (Line 49-51) of the Introduction section.
Point 3: Row 203 "reflects" implies that this is the only explanation, but this is not necessarily the case; better to use a more nuanced formulation such as "likely to reflect".
Response 3: Thanks for the advice of word choice. We followed your advice and used “likely to reflect" to replace “reflect”.
Point 4: Some paragraph in the discussion are not consequence of data analysis and could be dropped or made substantially synthetics (row 213-221; 274-286)
Response 4: Thanks for these great suggestions. We proofread the discussion part again and systematically improved these two sections.
Lines 213-221 have been revised as follows:
“This study showed a relatively high and stable prevalence of hypertension among middle-aged and elderly Chinese, with a down-ward turning point appeared in 2015. Meanwhile, it also reported an increased rate of hypertension awareness, treatment and control. The effective management of hypertension among the middle-aged and elderly Response: to the implementation of NPHI started in 2009, which aims at offering equitable essential public health services for all Chinese people, including hypertension and diabetes screening, standard management, and health education.”
We have dropped the following sentences (Lines 279-283):
“It would be effective if doctors' home visits and health education sessions were increased, access to affordable health services improved, and patients' adherence to medication treatment also improved. Moreover, the scope of universal health coverage should go beyond essential health services and financial protection to include more aspects of effective health coverage, such as actual health gain and healthcare outcomes.”
Paragraphs regarding Lines 274-286 have been further improved as follows:
“The reality of two-thirds of hypertensive patients experiencing uncontrolled blood pressure is likely to be the result of inconsistent, inadequate treatment or insufficient quality of health service provision. Though NPHI was developed to ensure optimal management of hypertension and diabetes for Chinese adults aged 35 and above, yet the inconsistent and uncoordinated primary care delivery system, lack of qualified physicians and public health workers, and absence of effective surveillance system handicapped the implementation of NPHI. A survey showed that over 60% of Chinese adults with hypertension were still not receiving public health services in 2015, that is, the lack of health education to population, including advice on weight control, exercise, healthy diet, and smoking control.”
“Major changes need to be made to improve the efficiency of service delivery for hypertension and substantially improve the effective control of hypertension in China. In response to this challenge, the current primary care system in China needs to be further strengthened and integrated with other reforms in public health service provision and the EMS. With comprehensive and continuous efforts in reforming the healthcare system, a more efficient and equitable health system would result, ensuring further improvement in the management of chronic diseases in China.”
Point 5: Table 2: the meaning of weighted and unweighted must be explicitly stated; also p values needs to be split for the two temporal intervals; similarly in table 3 and captions need clarification
Response 5: We added the related explanation in the Method section, line 129-132 as follows:
“Considering the potential systematic error caused by sampling design and missing values. both the weighted and unweighted proportions for key variables of hypertension management were reported in descriptive results, using the weighting codes created by the CHARLS team.”
And we improved the captions of the tables 2-3. The sentence for explaining “Unweighted percentage referred to the results of direct observation and weighted referred to the results after adjusting design effect.” were added the notes at the bottom of tables.
In addition, the weighted and unweighted results of statistical test (P values) are consistent, therefore we reported the p values based on weighted results, considering comments from reviewer 1. For clarification, we added a note below table 2 and table 3: “P values reported for intertemporal differences of the weighted proportion”.
Point 6: The graphic aspect of the figures 1 and 2 can be improved.
Response 6: We have further improved the paragraph related to the description of figures 1 and 2, as follows:
“In terms of hypertension management, the worst indicator of pro-rich inequality was the control of blood pressure (Figure 1). Regarding the health service use among hypertensive patients, compared with the outpatient service and health education access, the most unequal indicator was the inpatient visit, which was also more concentrated among wealthy individuals (Figure 2).”